# GuideGAN: Attention based spatial guidance for image-to-image translation

## Abstract

Recently, Generative Adversarial Network (GAN) and numbers of its variants have been widely used to solve the image-to-image translation problem and achieved extraordinary results in both a supervised and unsupervised manner. However, most GAN-based methods suffer from the imbalance problem between the generator and discriminator in practice. Namely, the relative model capacities of the generator and discriminator do not match, leading to mode collapse and/or diminished gradients. To tackle this problem, we propose a GuideGAN based on attention mechanism. More specifically, we arm the discriminator with an attention mechanism so not only it estimates the probability that its input is real, but also does it create an attention map that highlights the critical features for such prediction. This attention map then assists the generator to produce more plausible and realistic images. We extensively evaluate the proposed GuideGAN framework on a number of image transfer tasks. Both qualitative results and quantitative comparison demonstrate the superiority of our proposed approach.

## 1 Introduction

Generative Adversarial Networks (GANs) have drawn much attention during the past few years, due to their proven ability to generate realistic and sharp looking images. Various computer vision problems are solved using this framework, such as super-resolution (Ledig et al., 2017), colorization (Cao et al., 2017), denoising (Yang et al., 2018) and style transfer (Zhang et al., 2017). All these problems can be considered as an image-to-image translation problem, mapping an image from source domain to target domain, for instance, the super-resolution problem of trying to transfer a low-resolution image (source domain) to a corresponding high-resolution image (target domain). Existing literatures have shows that variants of GAN achieved impressive results in both a supervised and unsupervised setting. (Zhu et al., 2017; Liu et al., 2017; Wang et al., 2018; Isola et al., 2017; Choi et al., 2018; Huang et al., 2018)

Even with such great success, most GAN-based approaches are suffering from the imbalance between the generator and discriminator (Arjovsky & Bottou, 2017). In practice, the discriminator is usually too powerful for its task. Thus, the generator obtains very small gradients from discriminator and is hard to converge. Most state-of-the-art solutions are trying to find a new objective or add some new regularization terms to the cost function, which mainly affect the generator (Arjovsky et al., 2017; Arjovsky & Bottou, 2017; Mao et al., 2017; Nowozin et al., 2016; Zhang et al., 2018; Hu et al., 2018). To address this problem from a different direction, we want to borrow some power from the discriminator by incorporating the attention mechanism to help the generator. In this paper, we propose that the critical locating areas are more significant in the translation. The generator should pay more attention to some particular areas rather than the whole image.

Imagine a student is learning how to draw a horse. The standard discriminator, as a painting master, merely grades the student's painting and hopes that can help the student improve his work. On the other hand, another master will provide additional information. For instance, an error canvas circling each incorrect region. That is exactly our idea: we suggest that the student (generator) gains benefit from the second master (attention embedded discriminator). Our main contribution is threefold:

- **A flexible attention-augmented discriminator**: such discriminator provides not only the probability of realness, but an attention map from its perspective. Both *trainable attention module* and *post hoc attention* are implemented.
- **A unified GAN framework using attention information**: to improve the translation of the generator, we combine the attention map with raw input via two concatenate methods: 1) convert the input to a RGBA image by adding an alpha channel; 2) compute the residual Hadamard production of the attention map and corresponding original input, based on RAM; (Wang et al., 2017)
- **Extensive experiment validation on different benchmarks**: we provide extensive experimental validation of GuideGAN on different benchmarks; both the qualitative results and quantitative comparisons against state-of-the-art methods demonstrate the effectiveness of our approach.

To the best of our knowledge, we are the first to report image-to-image translation results using the attention information from discriminator. Different with previous approaches, our framework strengthens the communication and guidance between the generator and discriminator. At a high level, the significance of our work is also on discovering that the attention information from auxiliary network affects the result of image-to-image translation, which we think would be influential to other related research in the future.

## 2 RELATIVE WORKS

**Generative Adversarial Network** GANs have achieved impressive results in image translation tasks (Denton et al., 2015; Radford et al., 2015; Isola et al., 2017; Kim et al., 2017; Ledig et al., 2017). Typically, GAN consists of two components: a generator and a discriminator. The generator is trained to fool the discriminator which in turn tries to distinguish between real and synthesised samples. Various improvements to GANs have been proposed, like improved objective function (Mao et al., 2017; Arjovsky et al., 2017) and advanced training strategies (Gulrajani et al., 2017; Nowozin et al., 2016). A recent framework, *FAL* (Huh et al., 2019), iteratively improves the synthesized image using a spatial discriminator. However, they either didn't collect enough information from the discriminator or need more forward pass to stabilize the result.

**Image Translation** Image to image translation can be considered as a generative process conditioned on an input image. *pix2pix* (Isola et al., 2017) was the first unified framework for supervised image-to-image translation based on conditional GAN (cGAN) (Mirza & Osindero, 2014). *TextureGAN* (Xian et al., 2018) solves the sketch-to-image problem using user defined texture patch. Gonzalez-Garcia et al. (2018) adopted disentanglement representation to improve the rendering process and Tang et al. (2019) utilized the extra semantic information to guide the generation.

Despite the promising results they achieved, the above methods are generally not applicable in practice due to the lack of paired data. Several interesting frameworks have be proposed to solve this unsupervised image-to-image translation problem. Cycle consistency loss was first proposed in *CycleGAN* (Zhu et al., 2017) and is widely used by other unsupervised image translation frameworks. UNIT (Liu et al., 2017) improves the translation with shared latent space assumption, which is the fundamental of MUNIT (Huang et al., 2018) that handles multi-modal translation. In contrast, our flexible framework can be applied on both supervised and unsupervised settings.

**Attention Learning** Generally, attention can be viewed as guidance to bias the allocation of available processing resources towards the most informative components of an input. Contemporary approaches are divided into two categories: post hoc network analysis and trainable attention module. The former scheme has been predominantly employed to access network reasoning for the visual object recognition task (Simonyan et al., 2013; Zhou et al., 2016; Selvaraju et al., 2017; Chattopadhay et al., 2018). Trainable attention models fall into two main sub-categories, hard (stochastic) that requires reinforcement training and soft (deterministic) that can be trained end-to-end (Wang et al., 2017; Hu et al., 2018; Woo et al., 2018).

Attention is also widely used in image translation. Ma et al. (2018) proposed the DA-GAN framework, which learns a deep attention encoder to discover the instance level correspondences. Mejjati et al. (2018) separates the object and background using a trainable attention network. *InstaGAN* (Mo et al., 2018) incorporates the instance information, like segmentation masks, to improve the multi-instance transfiguraiton. Generally, these methods are trying to boost the attention embedded component, while we are using the attention mechanism to transfer more information from the

discriminator to the generator. We directly compare against several state-of-the-art approaches in Section 4.

## 3 METHOD

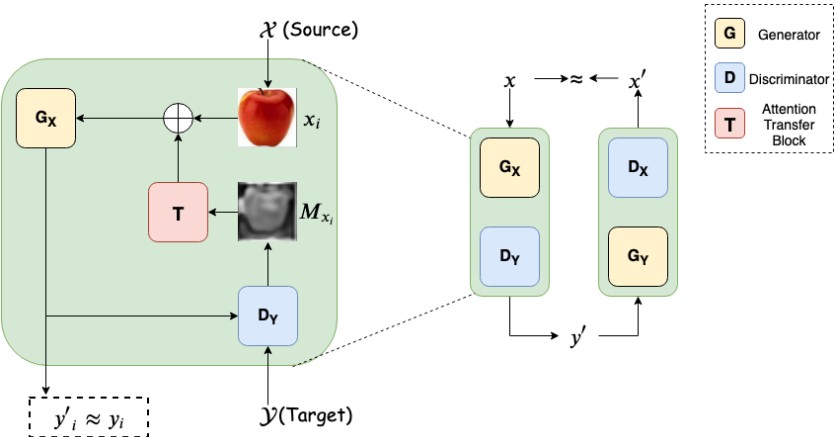

Figure 1: Overview of our framework. Left block is a standard GAN with an attention embedded discriminator. $M_x$ is the attention map provided by the discriminator. The L1 loss between generated $y_i'$ and corresponding ground truth $y_i$ is computed. Right side is the framework for unsupervised translation using cycle consistency. Ground truth $y_i$ is not available and the L1 loss between $x$ and $x'$ is calculated instead.

Consider images from two different domains, source domain $\mathcal{X}$ and target domain $\mathcal{Y}$. Data instances in source domain $x \in \mathcal{X}$ follow the distribution $P_x$, whereas instances in target domain $y \in \mathcal{Y}$ follow the distribution $P_y$. Notice that we do not have labels in both $\mathcal{X}$ and $\mathcal{Y}$. Our goal, in the problem setting of image-to-image translation, aims to learn mapping functions $Gs$ across these two different image domains, $G_X : x \to y$ and/or $G_Y : y \to x$, such that the differences between $P_x$ and $G_Y \circ P_y$ and the difference between $P_y$ and $G_X \circ P_x$ are minimized. From the perspective of statistics, learning those two mapping functions can also be formulated as estimating the conditional distribution $P(x|y)$ and $P(y|x)$.

The main and unique idea of our approach is to incorporate the attention map generated by the discriminator, *i.e.*, augment a space of attention information $A$ to the original input space $X$, to improve the image-to-image translation. The attention map can be further transformed to an extra alpha channel $\alpha$ (a mask channel with weight) or be interpreted as a pixel weight map. In this paper, different attention mechanisms and concatenation methods have been studied and achieve promising results based on a different task setting. Formally, our approach can be described as a joint-mapping learning from attention-augmented space $X \oplus A_X$ to $Y$, and $Y \oplus A_Y$ to $X$ if cycle consistency applied, where $\oplus$ is the concatenate operation. Our method explicitly forces the generator to put more processing resources to attended areas so it can conduct a sharp and clear translation. Generally, this approach can be applied to any conditional GAN-based translation, hence, we call it GuideGAN. We will present the detail of our approach in the following subsections.

### 3.1 ARCHITECTURE

Our framework, as illustrated in Figure 1, is built upon GAN and attention mechanism. For the supervised learning setting, it consists of three components, a generator $G_X$, a discriminator $D_Y$ and an attention transfer block $T$. It can be extended to unsupervised setting using CycleGAN framework easily, which now has five components: including two generators $G_X$ and $G_Y$, two domain adversarial discriminators $D_Y$ and $D_X$, and one shared attention transfer component $T$.

The training is based on each generator-discriminator pair. Considering a standard GAN, the generator $G_X$ translates an image $x_i$ in $\mathcal{X}$ to an image in domain $\mathcal{Y}$ and the discriminator $D_Y$ tries to

distinguish whether its input is a real or fake image in domain $\mathcal{Y}$. Here, we denote $y'_i = G_X(x_i)$ as the output of generator, given $x_i$. Our attention embedded discriminator not only returns the probability of realness, $D_Y(y'_i) \in [0, 1]$, but also an attention map $A_{x_i}$ that highlights the focusing area of $D_Y$. This attention map $A_{x_i}$ will be passed to the attention transfer block $T$ to create an alpha channel or a pixel weight map, depending on the concatenation method, which will be described in Section 3.3. For simplicity, the constructed term is denoted as $M_{x_i}$ given $A_{x_i}$, despite its actual interpolation. Noteworthy is the input of our generator $G_X$ is actually the concatenation of $x_i$ and $M_{x_i}$, which is represented as $x'_i = x_i \oplus M_{x_i}$. At the start of the training, the attention map of each image is not available so we initialize it as an all-ones matrix $A_{x_i} \in \mathcal{R}^{m \times n}$, where $m \times n$ is the shape of the input image. Other initialization methods, like random noise, have also be examined but have limited impact on the final result. The translation process of generator $G_X$ can be formulated as:

$$y'^{(k+1)}_i = G_X(x_i \oplus T(D(y'^{(k)}_i)); \theta), k = 0, 1, 2, \ldots \tag{1}$$

where $k$ and $k + 1$ denote the index of iteration and $\theta$ is the parameter of $G_X$. We emphasize that the attention map from $D_Y$ is crucial because it allows $G_X$ to focus on informative areas. For example, if we only give the generator the raw input, $G_X$ may waste its processing resources on some inessential locations and $D_Y$ will beat it easily. As a consequence, the loss of the discriminator quickly converges to zero and the generator can no longer efficiently update its parameter. Alternatively, by concatenating the raw input with $M_x$, the generator knows exactly where the discriminator is looking and allocates its processing resources properly on those areas. As illustrated in Figure 1, we can easily extend this framework to perform the unsupervised translation by adding another GAN component and enforcing cycle consistency.

## 3.2 Attention Map

Remember that our discriminator provides an extra attention map $A_{x_i}$ for each image generated from $x_i$. Therefore, we consider both *post hoc attention* (PHA) that does not change the capacity of the discriminator, and *trainable attention module* (TAM), which enhances the discriminator's distinguishing power.

Given input $x$, the post hoc attention map is constructed from the backward gradients, forward activation, or the mix of them. We use PatchGAN (Isola et al., 2017) as the bone of our discriminator. The network can be formulated as $D = \{l_0, l_1, \ldots, l_m\}$ where $l_i$ denotes $i$-th convolution layer in the network, and $Act_D = \{a_1, a_2, \ldots, a_m\}$ is the set of activation map of corresponding layer. This kind of attention map is sensitive to layer selection; different layer selection leads to different attention map (Mei et al., 2019). More specifically, if $t$ is the chosen layer, the attention map can be described as:

$$M = g(\frac{1}{c} \sum_{i=1}^{c} |a_{t,i}|) \tag{2}$$

where $c$ is the number of channels in $t$-th layer and $g(\cdot)$ applies the min-max normalization. This attention map only requires minor computation and works surprisingly well in most cases, but it may not achieve promising results when handling complex images. On the contrary, a TAM is suitable for such complex input since it simultaneously increases the capacity of generator and discriminator.

Our TAM follows the same 2-branch architecture of the attention block in RAM (Wang et al., 2017). See Appendix A for implementation details. However, each branch in their implementation contains several *ResBlock* He et al. (2016), which makes it impractical in our framework due to two reasons: 1) The discriminator conducts a simple binary classification 2) The capacity gap between generator and discriminator is already significant. We replaced the *Resblock* by a simple convolution layer to simplify the network structure. First few layers of the discriminator extract the low-level information of the input, and passes it to following branches. Given the trunk branch output $T(x)$ with the input $x$, the mask branch learns an attention map $M(x)$ that softly weights the output of trunk branch. Put these two outputs together:

$$E_{i,c} = (M_{i,c}(x) + 1) \times T_{i,c}(x) \tag{3}$$

where $i$ ranges over all spatial positions and $c \in \{1, 2, ..., C\}$ is the index of channels. Finally, a few consecutive convolutional layers will do the final prediction based on $E$ and attention map $\frac{1}{C} \sum_c M_c(x)$ constructed from mask branch output will be returned.

## 3.3 CONCATENATION

In this section, we propose two methods to blend the attention map $M_x$ with its corresponding input $x$. The first one is based on the aforementioned attention module in the RAM (Wang et al., 2017). We compute the Residual Hadamard Production (RHP) of the attention map and original input. The reason of this operation are 1) Dot production with the attention range from zero to one will degrade the pixel value and cause fractional pixel problem (Mejjati et al., 2018), 2) Attention mask can potentially break good property of the raw input. For example, some pixels are not crucial in distinguish real and fake image, but they are still important for the image translation process. This RHP can be formulated as:

$$x' = (g(M_x; \phi) + 1) \times x \tag{4}$$

where $g(; \phi)$ is a transfer function that up-sample the attention map to the shape of original input. Another more intuitive concatenation is converting an RGB image to its RGBA version. RGBA, as a color space, stands for red-green-blue-alpha. It is the three-channel RGB color model supplemented with a $4$-th alpha channel that indicates how opaque each pixel is. This concatenation somehow makes nonessential areas more transparent thus highlighting the crucial locations. Formally, it is described as:

$$x' = \{x_r, x_g, x_a, g(M_x; \phi)\} \tag{5}$$

where $g(; \phi)$ is a transfer function that maps attention map to alpha channel. Follow the standard image pre-processing step, this concatenation can also be applied on gray scale image. Gray scale image can be transformed into RGB image by repeating its intensity for each RGB channel.

## 3.4 TRAINING LOSS

Let's start with supervised translation. The adversarial loss of the generator $G$ and its discriminator $D$ can be expressed as:

$$L_{GAN}(G, D) = \mathbb{E}_{y \sim p_{data}(y)}[\log D(y)] + \mathbb{E}_{x \sim p_{data}(x)}[\log(1 - D(G(x \oplus M_x)))] \tag{6}$$

which is the adversarial loss of vanilla GAN. $G$ aims to minimize this objective while an adversary D tries to maximize it, $i.e.$, $\min_G \max_D L_{GAN}(G, D)$. However, this cost function is well known for its training difficulty. We adopt the modified least-squares loss, proposed in LSGAN (Mao et al., 2017), to further stabilize the training process and improve the quality of generated images. The adversarial loss now becomes:

$$L_{GAN}(G, D) = \mathbb{E}_{y \sim p_{data}(y)}[(D(y) - 1)^2] + \mathbb{E}_{x \sim p_{data}(x)}[(G(x \oplus M_x))^2] \tag{7}$$

Adversarial loss alone does not guarantee a sound translation. It is beneficial to mix traditional loss like L1 distance or L2 distance between synthesized image and ground truth. Based on the suggestion from pix2pix (Isola et al., 2017) that L1 loss encourages less blurry, we chose L1 loss as part of our training objective:

$$L_{L1}(G) = \mathbb{E}_{x,y}[\|y - G(x')\|_1] \tag{8}$$

The final objective function in this setting is:

$$G^* = \arg \min_G \max_D L_{GAN}(G, D) + \lambda L_{L1}(G) \tag{9}$$

We can easily extend this framework to conduct unsupervised translation by adding another pair of generator and discriminator and enforcing cycle consistency. Assume the generator $G_X$ simulates

| Method | (A)pple↔(O)range | | (S)ummer↔(W)inter | | (A)pple↔(O)range | | (S)ummer↔(W)inter | |
|---|---|---|---|---|---|---|---|---|
| | A→O | O→A | S→W | W→S | A→O | O→A | S→W | W→S |
| PHA+Alpha | $7.25 \pm 0.83$ | $3.69 \pm 0.41$ | $1.86 \pm 0.24$ | $\mathbf{1.01 \pm 0.23}$ | $4.02 \pm 0.37$ | $\mathbf{4.11 \pm 0.31}$ | $1.04 \pm 0.12$ | $\mathbf{1.23 \pm 0.12}$ |
| PHA+RHP | $\mathbf{6.31 \pm 0.60}$ | $\mathbf{2.99 \pm 0.38}$ | $1.98 \pm 0.33$ | $1.03 \pm 0.26$ | $\mathbf{3.69 \pm 0.27}$ | $4.30 \pm 0.31$ | $1.18 \pm 0.16$ | $1.55 \pm 0.13$ |
| TAM+Alpha | $10.80 \pm 0.71$ | $7.26 \pm 0.47$ | $2.37 \pm 0.35$ | $1.76 \pm 0.37$ | $5.93 \pm 0.31$ | $6.70 \pm 0.36$ | $1.45 \pm 0.18$ | $1.71 \pm 0.15$ |
| TAM+RHP | $10.06 \pm 0.64$ | $6.81 \pm 0.45$ | $\mathbf{1.34 \pm 0.29}$ | $1.73 \pm 0.30$ | $5.54 \pm 0.31$ | $6.47 \pm 0.37$ | $\mathbf{0.82 \pm 0.14}$ | $1.72 \pm 0.15$ |

Table 1: KID×100 ± std.×100 (Lower is better) computed for different combination on *apple2orange* and *summer2winter*. Left 4 columns shown the target only KID and the rest 4 columns shown the mean KID (Lower the better). Best results are bolded.

the map function $G : X \rightarrow Y$ and discriminator $D_Y$ are trying to distinguish between $G(x)$ and $y$, the objective of this GAN component is $L_{GAN}(G_X, D_Y)$. The generator $G_Y$ and discriminator $D_X$ is doing the same task in an opposite direction, their loss function is $L_{GAN}(G_Y, D_X)$. Cycle consistency is employed in such unsupervised setting because it alleviate the shortness of paired data. It assumes that if a image $x$ from domain $\mathcal{X}$ has be translated to a fake image $\hat{y}$ in domain $\mathcal{Y}$, we should get the same image $x$ by applying $G_Y : Y \rightarrow X$. This behavior is formally presented as:

$$L_{cyc}(G_X, G_Y) = \mathbb{E}_{x \sim p_{data}(x)}[\|G_Y(G_X(x')) - x\|_1] + \mathbb{E}_{y \sim p_{data}(y)}[\|G_X(G_Y(y')) - y\|_1] \quad (10)$$

The final objective function for the unsupervised translation is:

$$G_X^*, G_Y^* = \arg \min_{G_X, G_Y} \max_{D_X, D_Y} L_{GAN}(G_X, D_Y) + L_{GAN}(G_Y, G_X) + \lambda L_{cyc}(G_X, G_Y) \quad (11)$$

## 4 EXPERIMENTS

A crucial point of our framework is how we perform the inference in test phase. The attention map of same input from previous iterations can be used at training phase. However, this information is not available in testing, and some placeholders are required. To alleviate the problem that lead to this phenomenon, the generator should not rely too much on the attention map. Our proposed concatenation methods naturally handle this problem, since the attention map can only amplify the information but never hurt the original input. An all-one attention map is used as the placeholder based on the assumption that whole image is important.

### 4.1 ATTENTION AND CONCATENATION

Recall that we implemented two attention mechanisms and two concatenations. Then the problem is how to combine them properly. First, qualitative results are presented in Figure 2. As discussed in Section 3, TAM is not good at handling simple datasets, e.g. *apple2orange* while the results are more attractive for more complex *summer2winter* dataset. By comparing alpha concatenation with RHP concatenation under post hoc attention, we find that the contrast ratio of the synthesized image is normally too high and makes the image look unrealistic.

We also present a quantitative evaluation for each method combination in Table 1. Kernel Inception Distance (KID) (Bińkowski et al., 2018) is used as evaluation metric, which computes the squared MMD (Maximum Mean Discrepancy) between feature representations of real and generated images. Different from the Frchet Inception Distance (Heusel et al., 2017), KID is more reliable because of the unbiased estimator. While KID is unbounded, the lower its value, the more shared visual similarities there are between real and generated images. Numeric results in the table justified our previous observations. Based on the overall performance across different tasks, most experiments are using PHA and RHP by default.

### 4.2 OBJECT AND SCENERY TRANSLATION

We first evaluate our method on four benchmark datasets. *orange2apple*, *horse2zebra* (Deng et al., 2009) are for object transfer and *day2night*, *summer2winter* (Zhu et al., 2017) are two challenging scenery transfer tasks. *day2night* is cropped from BDD110k (Yu et al., 2018), which contains 7870

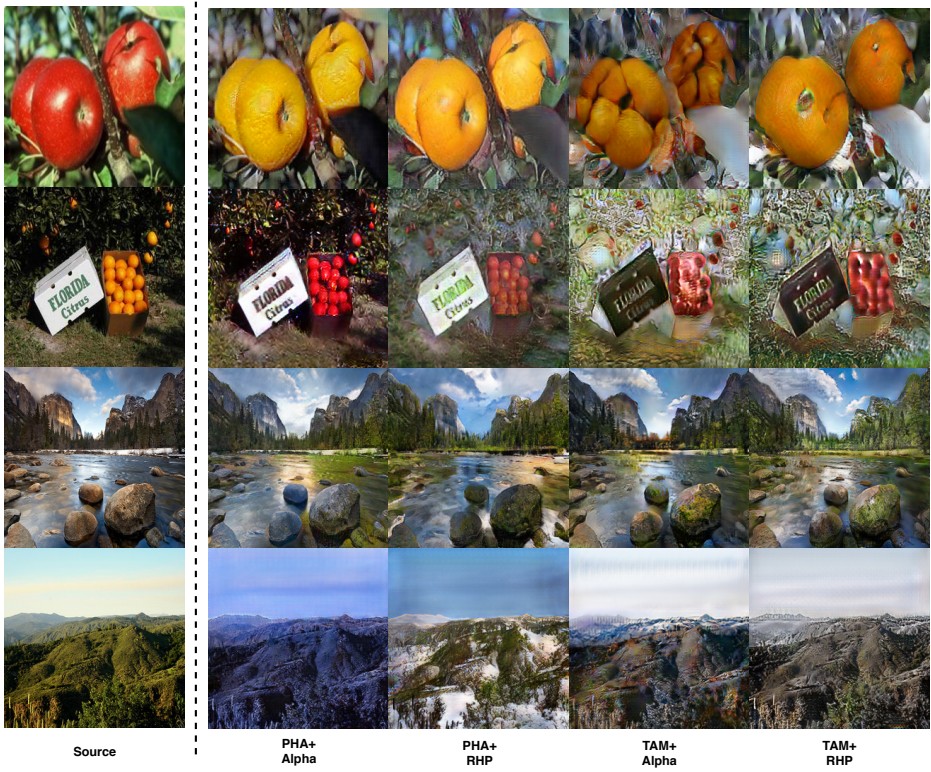

Figure 2: Different combination of attention and concatenation for *apple2orange* and *summer2winter*. First column is the real input. From second column to the right: PHA and alpha channel, PHA and RHP, TAM and alpha channel, TAM and RHP

| Method | (A)pple↔(O)range | | (H)orse↔(Z)ebra | | (D)ay↔(N)ight | | (S)ummer↔(W)inter | |
|---|---|---|---|---|---|---|---|---|
| | A→O | O→A | H→Z | Z→H | D→N | N→D | S→W | W→S |
| CycleGAN | 8.48 ± 0.53 | 5.94 ± 0.65 | 3.94 ± 0.41 | 4.87 ± 0.52 | **2.63 ± 0.20** | 7.68 ± 0.35 | 2.78 ± 0.22 | 1.86 ± 0.26 |
| StarGAN | 13.32 ± 0.52 | 11.19 ± 0.51 | 12.42 ± 0.74 | 12.21 ± 0.89 | 5.37 ± 0.43 | 8.49 ± 0.34 | 8.05 ± 0.37 | 8.72 ± 0.47 |
| AGGAN | 10.61 ± 0.79 | 4.57 ± 0.30 | 4.12 ± 0.80 | 4.46 ± 0.40 | 8.09 ± 0.37 | 7.85 ± 0.29 | 3.45 ± 0.43 | 2.75 ± 0.20 |
| UNIT | 17.41 ± 1.13 | 7.26 ± 0.57 | 12.25 ± 0.74 | 12.37 ± 0.84 | 2.83 ± 0.30 | 11.00 ± 0.53 | 6.20 ± 0.25 | 5.99 ± 0.28 |
| Ours (PHA+RHP) | **6.31 ± 0.60** | **2.99 ± 0.38** | **1.03 ± 0.35** | **3.42 ± 0.51** | 2.76 ± 0.32 | **6.96 ± 0.38** | **1.98 ± 0.33** | **1.03 ± 0.26** |

Table 2: KID×100 ± std.×100 (Lower is better) computed using only target domain for different methods and on different datasets. Best results are bolded.

images of daytime street traffic signs and 8592 night street traffic signs. All data were split into train and test randomly (80%/20% split).

Then for all tasks, we present target KID in Table 2 and mean KID in Table 3. The target only KID is meaningful when the background of an image is not important. For example, in *apple2orange* and *horse2zebra* tasks, we only care about objects in those images. Under such scenarios, our proposed framework outperforms all baselines in all tasks except one task in *day2night*. Still, our result is very close to the best. This observation is consistent with our qualitative evaluation in Figure 3, where our fake horse (zebra) is much more realistic than the counterparts produced by baselines. However, we can clearly see the background changed using our method, even though we still have leading performance on *apple2orange* and *summer2winter* regarding the mean KID between source and target domain. It's surprising to see that simplest CycleGAN model got the first place in *day2night*, which is a very hard task compared to two object transfer datasets. Another interesting observation is our framework got 6.96 for the task night→day, while CycleGAN got 7.68, given CycleGAN outperforms our method in the easier *day→night* direction. That's in the opposite direction as presented in Figre 4.

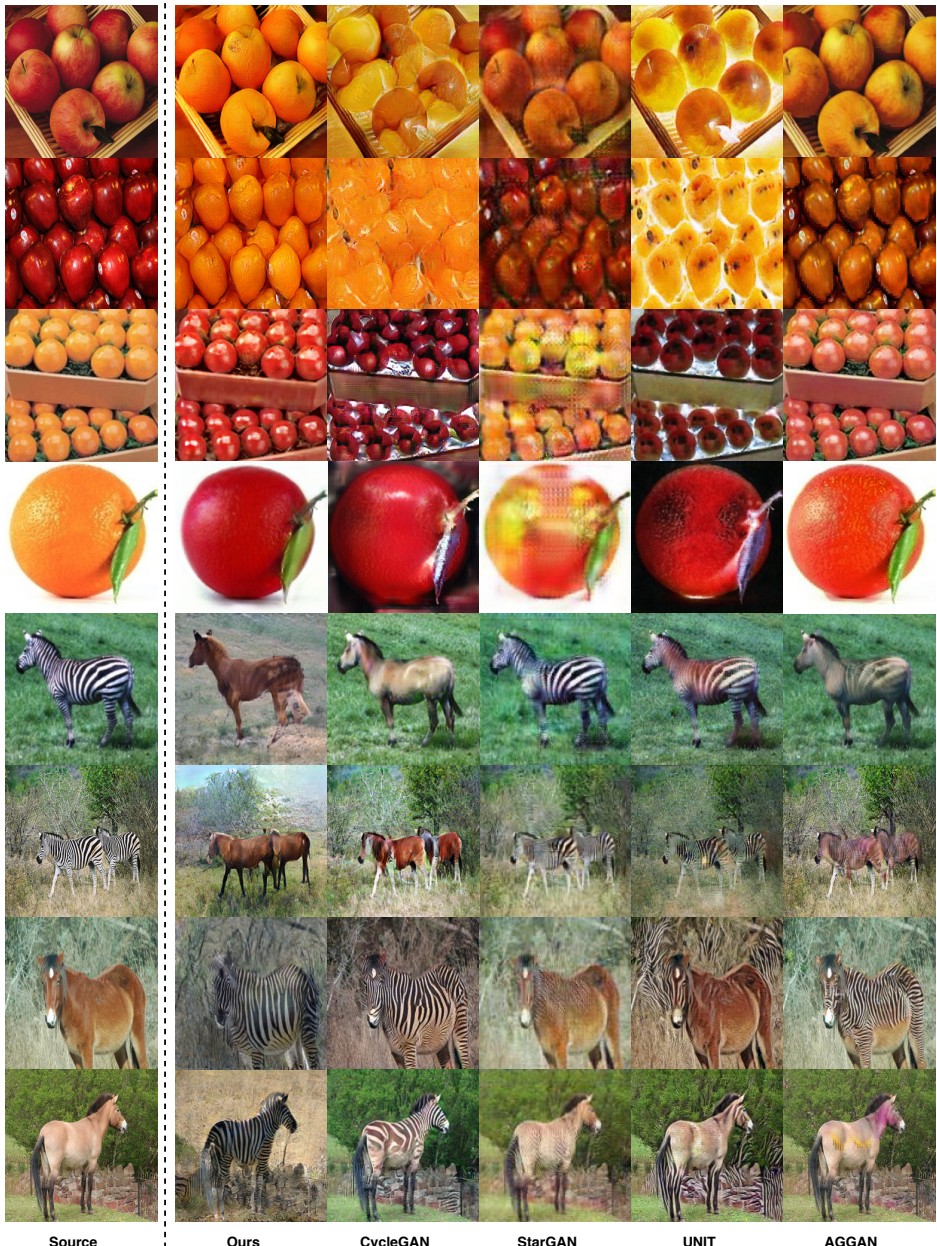

Figure 3: Image-to-Image translation results generated by different approaches on *apple2orange* and *horse2zebra*. Every two rows from top: *apple→orange*, *orange→apple*, *zebra→horse*, *horse→zebra*.

## 4.3 CITYSCAPE TRANSLATION

We then evaluate our method on *Cityscape* (Cordts et al., 2016) using FCN scores. Appendix B is shown for detailed evaluation protocol. We train photo→label and label→photo tasks on the *Cityscape*, and compare the output label images with the ground truth. We used only RHP concatenation for this task.

We find that our method significantly outperforms the baselines in these experiment, especially when PHA and RHP work together, as shown in Table 4. The image translation result is also presented in Figure 6. The significant improvement in the pixel-level accuracy comes from the guidance of the attention map, which aligns with our expectations. However, the improvement for metrics,

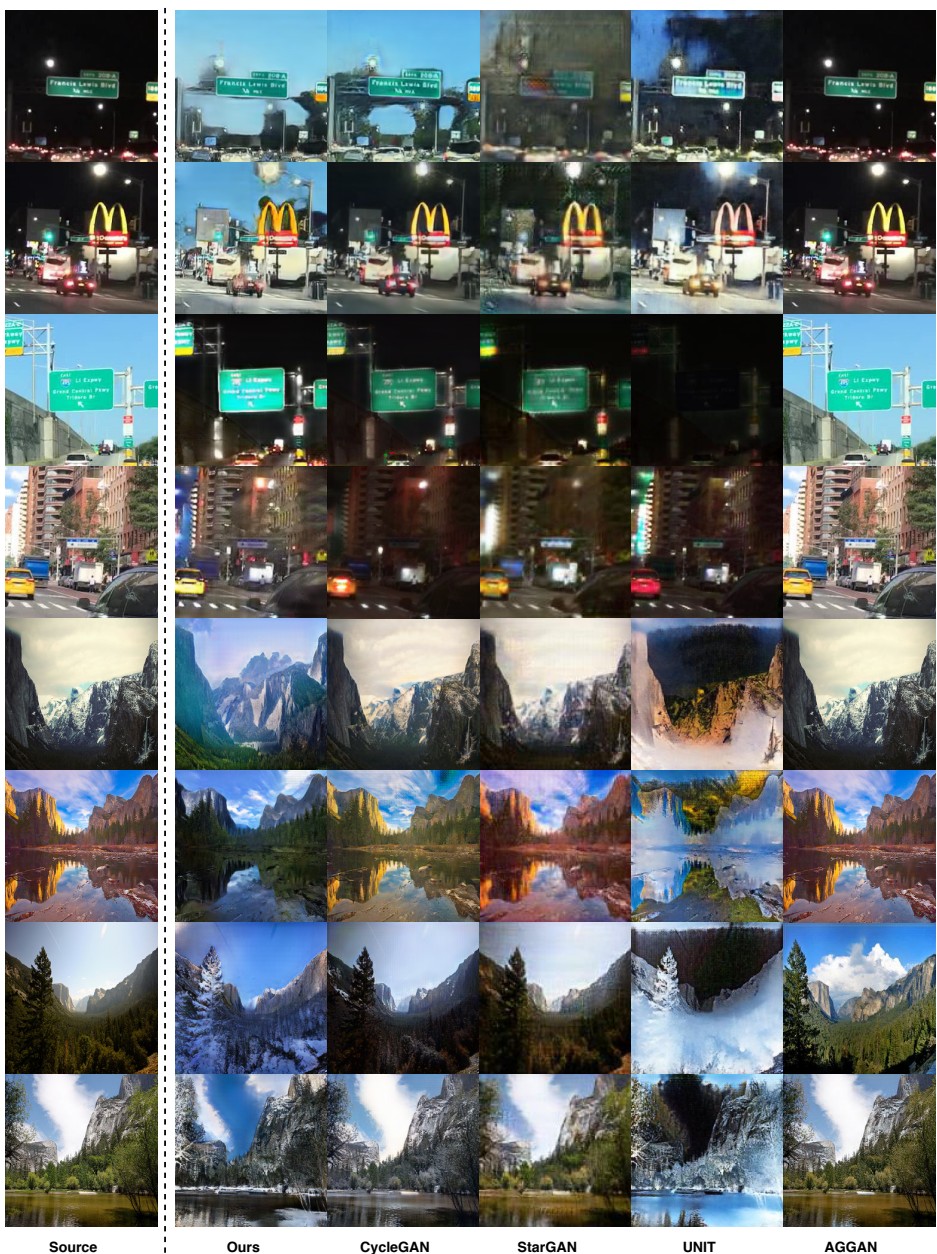

Figure 4: Image-to-Image translation results generated by different approaches on *day2night* and *summer2winter*. Every two rows from top: *night→day*, *day→night*, *winter→summer*, *summer→winter*.

| Method | (A)pple↔(O)range | | (H)orse↔(Z)ebra | | (D)ay↔(N)ight | | (S)ummer↔(W)inter | |
|---|---|---|---|---|---|---|---|---|
| | A→O | O→A | H→Z | Z→H | D→N | N→D | S→W | W→S |
| CycleGAN | 11.02 ± 0.60 | 9.82 ± 0.51 | 10.25 ± 0.25 | 11.44 ± 0.38 | **1.95 ± 0.13** | **3.63 ± 0.20** | 2.05 ± 0.12 | 3.34 ± 0.12 |
| StarGAN | 9.15 ± 0.43 | 8.31 ± 0.48 | 7.14 ± 0.48 | 4.50 ± 0.36 | 3.43 ± 0.20 | 5.18 ± 0.23 | 3.95 ± 0.17 | 4.14 ± 0.21 |
| AGGAN | 6.44 ± 0.69 | 5.32 ± 0.48 | 6.93 ± 0.27 | 6.71 ± 0.27 | 4.14 ± 0.14 | 4.97 ± 0.18 | 3.15 ± 0.19 | 2.45 ± 0.13 |
| UNIT | 11.68 ± 0.43 | 10.48 ± 0.67 | **4.91 ± 0.36** | **4.39 ± 0.33** | 2.48 ± 0.16 | 6.12 ± 0.29 | 3.51 ± 0.15 | 2.83 ± 0.12 |
| Ours (PHA+RHP) | **3.69 ± 0.27** | **4.30 ± 0.31** | 8.42 ± 0.47 | 8.46 ± 0.41 | 2.48 ± 0.15 | 4.58 ± 0.23 | **1.18 ± 0.16** | **1.55 ± 0.13** |

Table 3: KID×100 ± std.×100 (Lower is better) computed using both target and source domain for different methods and on different datasets. Best results are bolded.

such as class accuracy and Intersection over Union (IoU), are limited. It is probably because the attention map only focuses on a few domain specific classes, so the generator works too hard on those classes and ignores others. Small class number per image might be another potential reason

| Method | Label→Photo | | | Photo→Label | | |
|---|---|---|---|---|---|---|
| | Per-pixel acc. | Per-class acc. | IoU | Per-pixel acc. | Per-class acc. | IoU |
| CycleGAN | 0.42 | 0.15 | 0.10 | 0.56 | 0.21 | 0.17 |
| UNIT | 0.48 | 0.17 | 0.11 | 0.58 | 0.18 | 0.14 |
| AGGAN | 0.37 | 0.11 | 0.09 | 0.49 | 0.14 | 0.10 |
| StarGAN | 0.47 | 0.16 | 0.11 | **0.61** | 0.21 | 0.17 |
| Ours (PHA) | **0.52** | **0.20** | **0.12** | 0.60 | **0.24** | **0.19** |
| Ours (TAM) | 0.49 | 0.19 | 0.10 | 0.59 | 0.23 | **0.19** |

Table 4: FCN-scores (Higher is better) for different methods, evaluated on Cityscape label↔photos in unsupervised setting.

| Method | Label→Photo | | | Photo→Label | | |
|---|---|---|---|---|---|---|
| | Per-pixel acc. | Per-class acc. | IoU | Per-pixel acc. | Per-class acc. | IoU |
| GAN | 0.22 | 0.05 | 0.01 | 0.32 | 0.08 | 0.02 |
| cGAN | 0.57 | 0.20 | 0.14 | 0.71 | 0.26 | 0.21 |
| pix2pix | 0.61 | 0.22 | **0.16** | 0.80 | **0.43** | **0.32** |
| Ours(PHA) | **0.63** | **0.23** | **0.16** | **0.81** | 0.42 | **0.32** |
| Ours(TAM) | **0.63** | 0.22 | **0.16** | 0.75 | 0.40 | 0.30 |

Table 5: FCN-scores (Higher is better) for different methods, evaluated on Cityscape label↔photos in supervised setting. See Appendix C for qualitative result.

for this phenomenon, since we cannot increase the accuracy for nonexistent class objects. Empirical justifications are available in Appendix D.

Meanwhile, the improvement of the supervised translation is not as sharp as the unsupervised translation according to Table 5. Yet it still shows that we can further improve the translation results with little extra computation, especially when PHA has been chosen. We believe that the major reason actually due to strong regularizations, which are from the L1 distance between paired images. The generator receives two feedbacks when paired image is available. 1) The L1 loss between paired image and 2) The prediction from the discriminator. Recall that the idea behind our framework is letting the discriminator provide more useful information, but maybe the information from L1 loss is already sufficient. Appendix C offers qualitative result.

## 5 CONCLUSION

we have proposed a novel method incorporating attention map from discriminator for image-to-image translation. The experiments on different datasets have shown successful translation in both supervised and unsupervised setting. We remark that our idea can apply on any GAN-based model with little modification, such as those baselines in the paper. Nonetheless, the results are sensitive to the selection of attention module and concatenation. Investigating the impact of different attention mechanism and new tasks could be an interesting research direction in the future.

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

## A    IMPLEMENTATION DETAILS

For the unsupervised *Cityscape* translation, we adopted the network architectures of CycleGAN (Zhu et al., 2017) as the basic of our proposed model. In specific, we adopted the ResNet 6-blocks (He et al., 2016) generator and the PatchGAN (Isola et al., 2017) discriminator. This generator contains 2 down-sampling blocks, 6 residual blocks and 2 up-sampling blocks. For the supervised translation, we adopted the UNet-128 (Ronneberger et al., 2015) generator and a same PatchGAN discriminator. The PatchGAN discriminator is composed of 5 convolution layers, including normalization and ReLU layers.

Before diving into the detail of our modified discriminator, let us first describe the details of RAM's 2-branch architecture (Wang et al., 2017). They built a very deep network with numbers of *attention blocks*. Each *attention block* contains two branches: mask branch and trunk branch. Mask branch cascades the input features through a bottom-up top-down architecture that mimics human attention. Trunk branch is applied as feature processing. To build a TAM discriminator with this 2-branch architecture, we replaced the *ResBlock* by a simple convolution layer, as presented in the left part of Figure 5. In this TAM discriminator, we use the first convolution layer as feature extractor, three consecutive convolution layers for trunk branch and the last one convolution layer for classifier. The mask branch is composed of two downsampling layers, two convolution layers and one upsampling layer.

As presented in the right side of Figure 5, we selected the 4-th convolution layer to compute the post hoc attention map, based on the formula in Section 3.2. All attention maps will be detached from the computation graph and be resized to the shape of original input. Either by a bilinear upsampling layer or by a small 3-layer neural network.

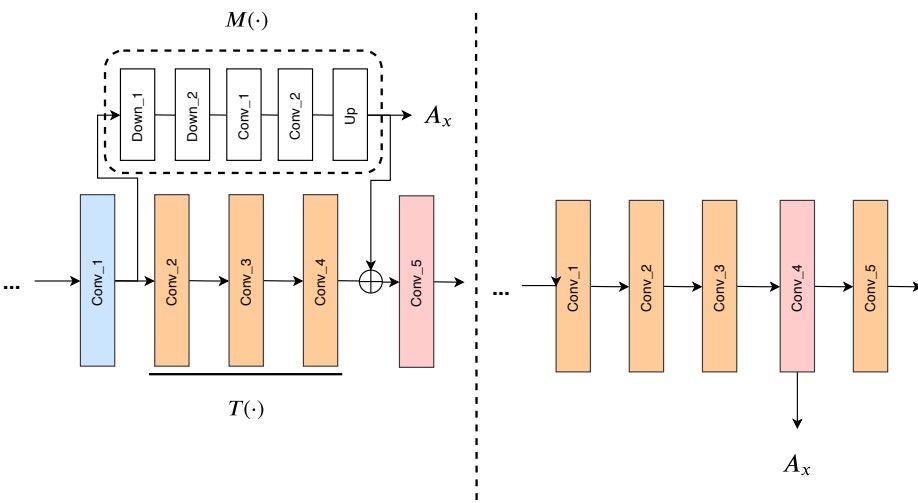

Figure 5: Left: PatchGAN discriminator using TAM, the attention map is denoted as $A_x$; Right: Patch discriminator using post hoc attention, the attention map $A_x$ is computed from 4-th conv layer.

*horse2zebra*, *apple2orange* and *day2night* tasks are performed under the unsupervised setting. For this three tasks, we adopted ResNet 9-blocks generator and aforementioned PatchGAN discriminator. Similar to prior works, we applied Instance Normalization (IN) for both generators and discriminators. In the preprocessing step, we resized the input image to $143 \times 143$ then randomly cropping back to $128 \times 128$ for all *Cityscape* related tasks. We resized the input image to $286 \times 286$ then randomly cropping back to $256 \times 256$ for the rest tasks.

For all the experiments, we simply set the weight factor of the GAN loss to 10 and the weight factor of L1 loss to 10 for our objective. For example, our implementation uses following objective for supervised training.

$$G^* = \arg \min_G \max_D 10 L_{GAN}(G, D) + 10 L_{L1}(G)$$

We used Adam optimizer with batch size 1, training on a Quadro 8000 GPU. All networks were trained from scratch, with learning rate of 0.0002 for both the generator and discriminator, and $\beta_1 = 0.5$, $\beta_2 = 0.999$ for the optimizer. Similar to CycleGAN, we kept learning rate for first 100 epochs and linearly decayed to 0 for next 100 epochs for *apple2orange* and Cityscape related tasks, and kept learning rate for first 50 epochs and linearly decayed to 0 for next 50 epochs for *horse2zebra* and *day2night* datasets.

## B    EVALUATION METRIC IN DETAIL

Evaluating the quality of synthesized images is an open and difficult problem. In this paper, we trained a network to perform *label→photo* and *photo→label* translation in both supervised and unsupervised manner. Classical metrics such L1, or L2, distance between the real image and synthesized image are not suitable since they do not assess joint statistics of the result. Researchers in image segmentation are widely using a pretrained semantic classifier to measure the discriminability of the generated image as a surrogate metric. The assumption behind such measurement is that if the generated images are indeed realistic, classifiers pretrained on real images should classify the synthesized image correctly as well. For the *Cityscapes* dataset, we used the FCN-8s (Long et al., 2015) network released by Zhu et al. (2017), which is pretrained on the *Cityscape* dataset.

The metrics we used in our experiment are per-pixel accuracy, per-class accuracy. and Intersection over Union (IoU). Per-pixel accuracy, namely, is the ratio between the number of correctly predicted pixels and total number of pixels. It can be presented as:

$$\text{per-pix acc.} = \frac{P(x)}{M \times N} \tag{12}$$

where $P(x)$ denotes the number of correctly predicted pixels and $M \times N$ is the sharp of input image.

Per-class accuracy, also known as macro-average, is self explanatory. It computes the the accuracy for each class and then compute the average. It's formulated as:

$$\text{per-class acc.} = \frac{1}{|K|} \sum_{k \in K} \frac{P(x, k)}{G(k)} \tag{13}$$

where $K$ is the set of classes, $P(x, k)$ denotes the number of correctly predicted pixels for class $k$ and $G(k)$ is the total number of pixels that belongs to class $k$ in ground truth.

Intersection over Union (IoU) is another often used metric for image segmentation. It computes the ratio between the number of pixels seat in the intersection between predicted segmentation mask and ground truth, and the union of them. Let $P(x)$ be the prediction and $GT(x)$ be the ground truth.

$$\text{IoU} = \frac{|P(x) \cap GT(x)|}{|P(x) \cup GT(x)|} \tag{14}$$

For all three aforementioned metric, the highest score is one, and the closer to one, the better.

## C    QUALITATIVE RESULT FOR *Cityscape*

*Cityscape* translation result is presented in Figure 6.

## D    HYPOTHESIS JUSTIFICATION ON *Cityscape*

In order to empirically justify our hypothesis for the limited improvement over per-class accuracy and IoU in Section 4. We conduct two additional experiments and show its result here. We first

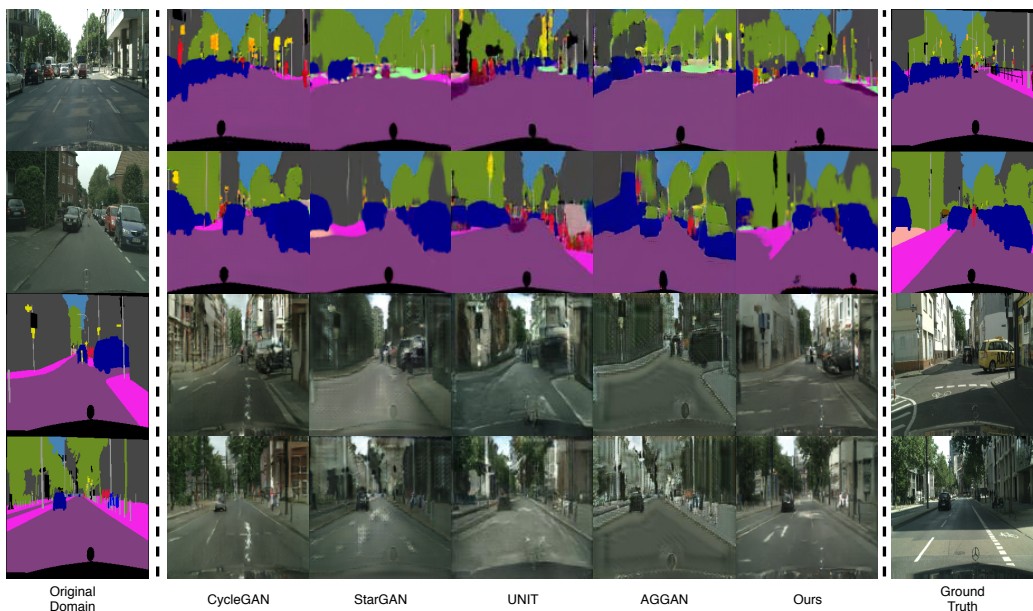

Figure 6: Different unsupervised translation methods for mapping labels↔photos trained on *Cityscape* images.

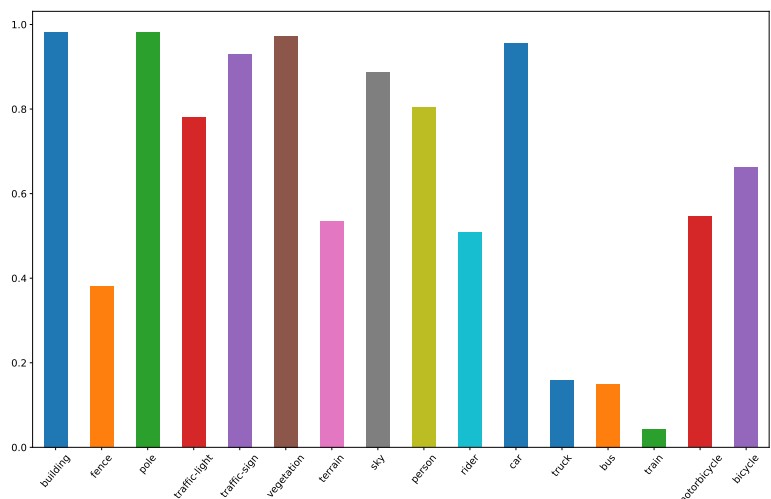

Figure 7: The statistic frequency for all 18 classes presented in the *Cityscape* dataset.

compute the statistic information of each class. Figure 7 shows the statistic frequency of each class. Namely, it tell us how many images contains the specific class in the dataset. Another useful statistic information, which tell us the average frequency for each class, is provided in Figure 8.

This statistic justified our second hypothesis that some class objects merely presented in the image thus it's hard to improve the per-class accuracy and IoU. We then extracted the attention map of whole training set and computed the average per-class attention map intensity. More specific, we first perform a binary normalization over all attention map using a threshold $\alpha$ (In this experiment we use $\alpha = 0.5$). So we assume a pixel is crucial if the attention value on it is larger than $\alpha$. For a specific class in one image, it's regarded as attended if at least half of its pixel is crucial. In the Figure 9, we show the average per-class attention map intensity in different epochs.

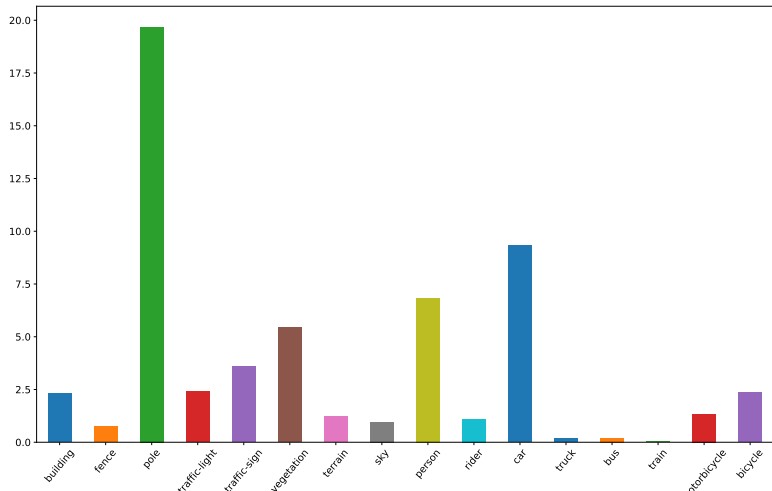

Figure 8: The average frequency for all 18 classes presented in the *Cityscape* dataset.

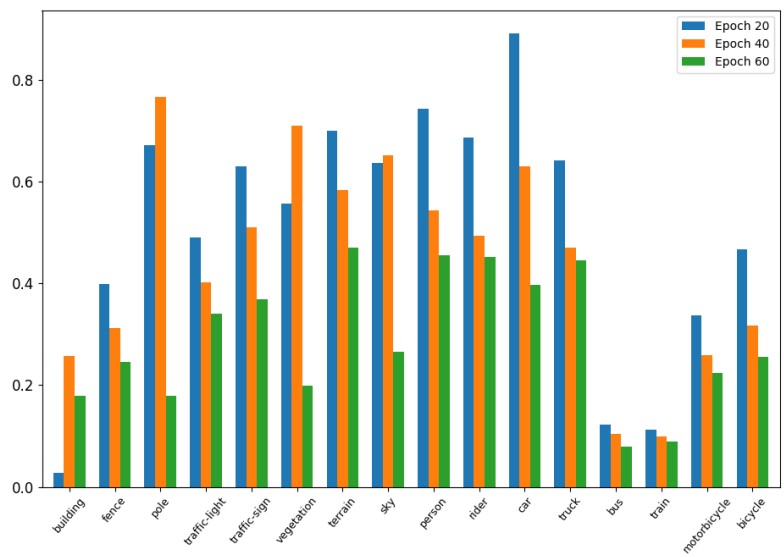

Figure 9: The average per-class attention map intensity in epochs 20, 40, 60.

Based on aforementioned figures, it's not hard to find out that the attention map are focusing on small classes. For example, rider and terrain. If the generator tried too hard to fix those small wholes but ignore the major classes, like car, the per-class accuracy and IoU will also be affected. Since the contribution from generating good riders and terrain is significantly less than the contribution from generating good cars. This experiment also justified our first hypothesis.

# E    ATTENTION MAP DURING TRAINING

We present some intermediate training results with its attention map in Figure 10, Figure 11 and Figure 12. The white area in the attention map indicates that region is important. Please note that the attention map indicates the behavior of the discriminator thus some of them may not make sense from human's perspective.

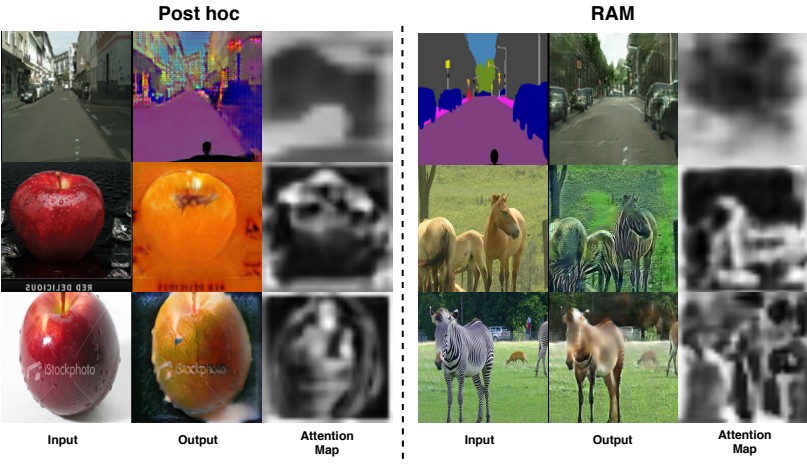

Figure 10: Inputs, outputs and corresponding attention maps at training epoch 10. Left: attention map generated by the post hoc attention; Right: attention map generated by RAM attention mechanism.

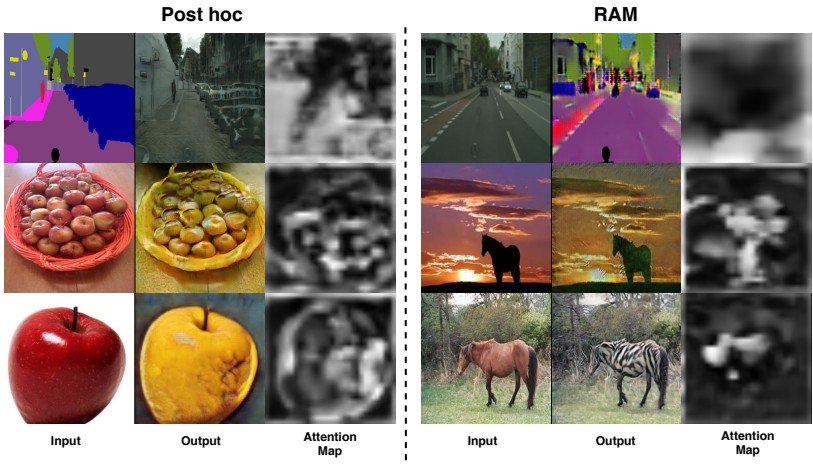

Figure 11: Inputs, outputs and corresponding attention maps at training epoch 50. Left: attention map generated by the post hoc attention; Right: attention map generated by RAM attention mechanism.

## F   MORE TRANSLATION RESULTS

More translation results are provided in this section.

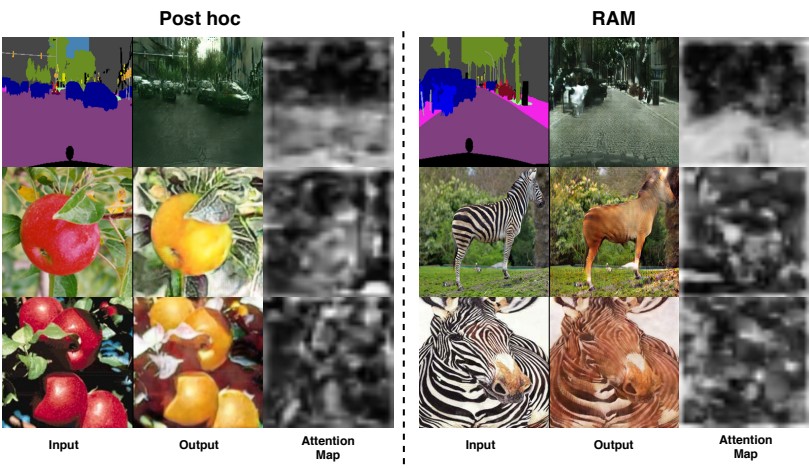

Figure 12: Inputs, outputs and corresponding attention maps at training epoch 100. Left: attention map generated by the post hoc attention; Right: attention map generated by RAM attention mechanism.

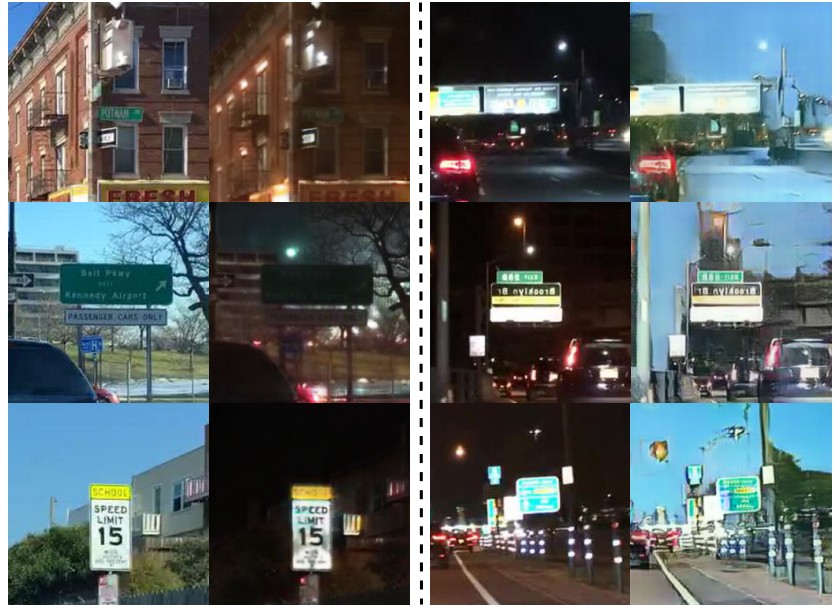

Figure 13: Additional translation results on *day2night* dataset with default setting (PHA + RHP). From left to right: real daytime images, fake night images, real night images, fake daytime images.

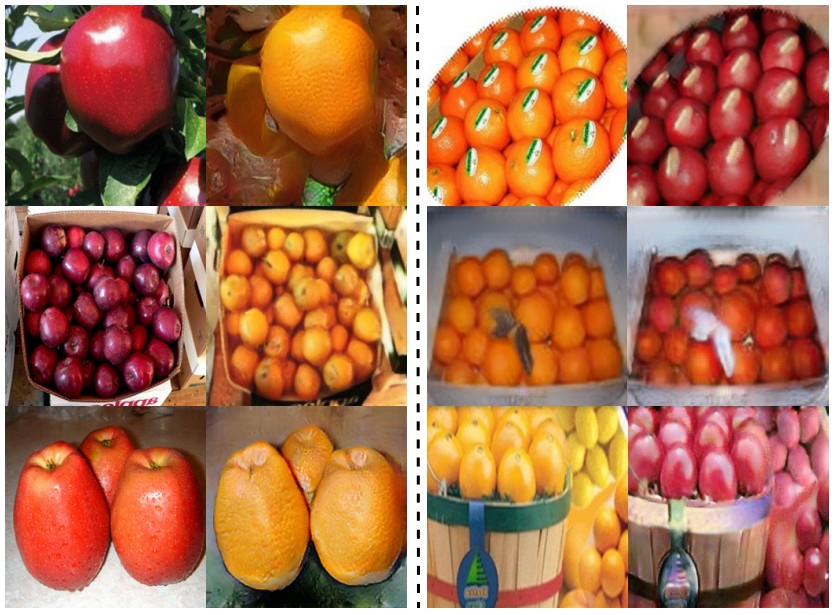

Figure 14: Additional translation results on *apple2orange* dataset with default setting (PHA + RHP). From left to right: real apple images, fake orange images, real orange images, fake apple images.

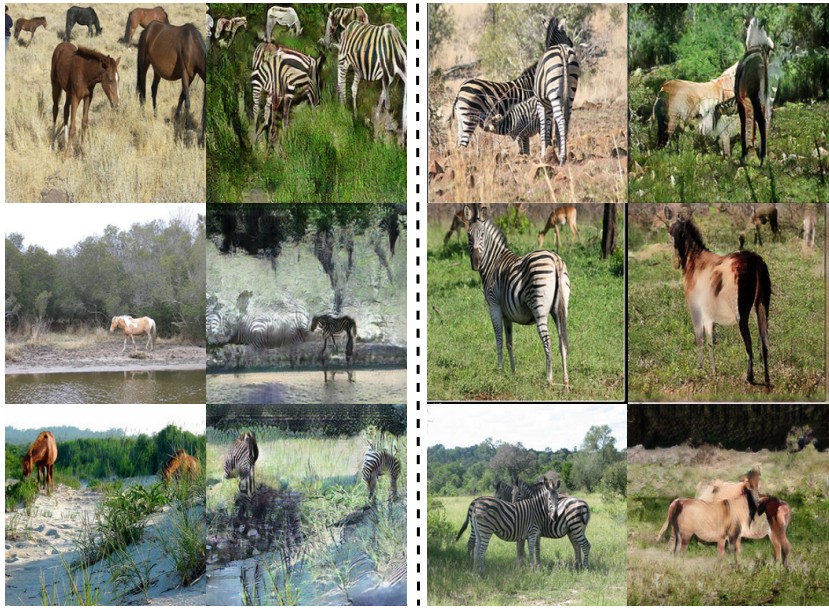

Figure 15: Additional translation results on *horse2zebra* dataset using RAM + RHP. From left to right: real horse images, fake zebra images, real zebra images, fake horse images.

