# OpenReview forum: "GUIDEGAN:  ATTENTION  BASED  SPATIAL  GUIDANCE FOR  IMAGE-TO-IMAGE TRANSLATION"
_ICLR.cc/2020/Conference — Reject_

### Official Review · AnonReviewer2 · 2019-10-21
**Official Blind Review #2**

**Rating:** 3

**Review:**

This paper introduces a feedback mechanism in the GAN framework which improves the quality of generated images in the context of image-to-image translation. The key contribution is that the discriminator not only predicts the probability of an image being real or fake, but also outputs a map which indicates where the generator should focus in the next iteration in order to make its results more convincing. The paper explores ways of obtaining such a map 1) by summing feature activations of the discriminator on a specific or group of layers  2) by predicting it via augmenting the capacity of the discriminator. After such a map is obtained, it is concatenated with the input image and fed iteratively to the generator.

The proposed setting have been tested on the setting of supervised and unsupervised image translation on 4 datasets. Quantitative experiments show that the proposed approach improves over other baselines. I think this paper introduces an interesting and important new GAN framework. However, I feel that the paper requires a major revision strengthening the experiments, before it can be reconsidered for ICLR:

More qualitative results showing comparisons with other algorithms should be shown for Day-Night, Apple-Orange, Horse-Zebra. The only comparison available in the paper is on the well constrained problem of segmentation maps to images.
More quantitative experiments should be provided for other datasets (perhaps using FID and KID). It is not entirely clear if the produced results are better than cycleGAN’s (my subjective analysis is that the results of cycleGAM look better on Fig. 2).
The paper is closely related to [Huh et al: Feedback Adversarial Learning: Spatial Feedback for Improving Generative Adversarial Networks] in that they share the idea of a feedback mechanism from the discriminator. It hence seems reasonable to compare with this approach.
I was struggling to understand precisely how the StarGAN results were obtained on CityScapes: As a multi-modal image-to-image translation model, StarGan takes as input, an image and a binary vector pointing to which modality to transform the image into. In the case of CityScape, there is no such multi-modality (at least none that is provided as ground truths, via for example, a binary vector). More details on this process would make the experimental section clearer.
Table 4.1 is often used however such a Table does not exist (probably Table 1,2 was meant here).


**Experience Assessment:**

I have published one or two papers in this area.

**Review Assessment: Checking Correctness Of Derivations And Theory:**

I assessed the sensibility of the derivations and theory.

**Review Assessment: Checking Correctness Of Experiments:**

I assessed the sensibility of the experiments.

**Review Assessment: Thoroughness In Paper Reading:**

I read the paper at least twice and used my best judgement in assessing the paper.

---

> ### Author Response · Authors · 2019-11-09
> **Response to Reviewer 2**
>
> Sorry for the late response and thank you for your constructive comments and demonstrated interest in the presented work, below we address each of your points in turn. 'G' denotes the generator and 'D' denotes the discriminator.
>
> (1). More experiments. We will add more comparison and KID/FID score in the later revision;
>
> (2). Comparison with [Huh et al: Feedback Adversarial Learning: Spatial Feedback for Improving Generative Adversarial Networks]. We have read this paper thoroughly and the main difference between their framework and ours is: they were trying to iteratively improve the synthesized image by predict some parameters, and use these parameters to linearly transform the learned hidden feature; Our method does not rely on such iterative mechanism, we let D provides an attention map that highlights crucial regions and guide G to focus on those regions. So we only need to conduct feedforward once during the test time. On the other hand, the discriminator in their framework outputs a response map instead of a probability, as stated by Reviewer 3, since G and D are playing a zero-sum game, D will try to hide the useful information in the response map. We also want to run their experiment but their Git repo is down for now.
>
> (3). Cityscape in StarGAN. The process of applying StarGAN on the Cityscape dataset is straightforward. As you said, we need a binary mask vector to control the translation. Assume $x$ comes from the photo domain and $y$ comes from the segmentation domain. We can use mask code [1, 0] for $x$ and [0, 1] for $y$. In this way, StarGAN becomes a multi-domain translation model. Do you think we need to explain the details of this in the paper?
>
> (4). We will fix the table reference in the later revision.

---

### Official Review · AnonReviewer1 · 2019-10-23
**Official Blind Review #1**

**Rating:** 3

**Review:**

[Summary]
This paper proposes a GAN with an attention-based discriminator for I2I translation, GuideGAN. The proposed discriminator provides the probability of real /fake and an attention map which reflects the salience for image generation. They apply their method to CycleGAN. GuideGAN is evaluated on the Cityscape dataset, horse2zibra, apple2orange, and day2night, compared to UNIT, CycleGAN, StarGAN, cGAN, and pix2pix.

[Pros]
- Quantitative results

[Cons]
- Novelty is restricted. Even if there is no work using attention in the discriminator, it is hard to tell that the use of attention is not novel.
- The used datasets are not challenging. Why the quantitative results are evaluated on the Cityscape? How about FID on other datasets such as CelebA and Summer2Winter, or higher-resolution data?
- The quantitative scores are not impressive. The gaps look insignificant.
- How large are additional parameters? and How long does it spend training compared to CycleGAN?
- The authors describe two concatenation methods. How about the results of simple RGBA?
- Why post hoc and RAM are evaluated on different datasets from each other in qualitative results?
- User study will be helpful.
- Basically, this architecture can be applied to various GANs. Do the authors have any result on other GANs?


[Minor]
In Figure 1, M is used without the definition.
In row 2 of page 3, follows --> follow


**Experience Assessment:**

I have published in this field for several years.

**Review Assessment: Checking Correctness Of Derivations And Theory:**

I assessed the sensibility of the derivations and theory.

**Review Assessment: Checking Correctness Of Experiments:**

I assessed the sensibility of the experiments.

**Review Assessment: Thoroughness In Paper Reading:**

I read the paper at least twice and used my best judgement in assessing the paper.

---

> ### Author Response · Authors · 2019-11-09
> **Response to Reviewer 1**
>
> Sorry for the late response and thank you for your constructive comments and demonstrated interest in the presented work, below we address each of your points in turn. 'G' denotes the generator and 'D' denotes the discriminator.
>
> (1). Novelty issue. Indeed, apply the existing attention mechanism in any network is not novel and we didn't intend to say that. The novelty of our framework is trying to let D provides more information since D is usually more powerful than G in practice. We were using attention mechanism to expose such extra information from D, and using this information to boost the performance of the generator;
>
> (2). Why we use Cityscape for quantitative results. We were just following previous literature and we will add some KID or FID score in the revised paper;
>
> (3). The quantitative scores are not impressive. We think the improvement for supervised I2I translation is indeed limited and we provide some justification, more empirical justification will be added in the later revision. However, the improvement over unsupervised I2I translation is significant. As stated in Answer (2), we will add some KID/FID scores in the revision.
>
> (4). Additional parameters and training time. Basically, our unsupervised I2I model is based on the official implementation of CycleGAN and the supervised version is built upon the pix2pix's official implementation. We didn't change G's architecture, except the input channel is 4 when using RGBA concatenation. For the post hoc attention, we didn't change D's architecture. For the trainable attention module, the modified discriminator (with mask & trunk branch) is powerful than the original one in the CycleGAN/pix2pix. We tried to replace the original discriminator with this modified D. But the performance of G decreases so we stay with the original one. This is expected because a more powerful D will further break the equilibrium between G and D. Both UNIT and StarGAN are more complex than CycleGAN so we think all of the baselines are comparable; The training time of CycleGAN is around 140s -170s per epoch and the training time of our framework is around 150s - 190s per epoch (For around 1000 256*256 images per epoch);
>
> (5). RGBA concatenation. In most cases, residual multiplication performs a little bit better than RGBA, so most experiment results presented are from residual multiplication. We will add more RGBA concatenation result in the revised paper;
>
> (6). Qualitative results with post hoc attention and RAM. We only show the best results from different combinations of concatenation and attention mechanisms. We can add more result and perform a comparison between different combination in the appendix;
>
> (7). We don't understand what kind of user study you are looking for. Could you please explain it in detail?
>
> (8). Results on other GANs. Based on Answer (4), our supervised model is built upon CycleGAN and the unsupervised model is based on pix2pix. They are two different GAN for different translation. We don't have results on other GAN framework but our method can be applied to other GANs easily since we don't need to change G's architecture in most cases. Only the activation from one specific layer of D is required, which also leaves D's architecture intact;
>
> (9). We will fix the typo and add the definition of M in the revision;

---

> > ### Comment · AnonReviewer1 · 2019-11-13
> > **For (7),**
> >
> > I am sorry for my late reply.
> > For your rebuttal, here I focus on the item (7) in the response.
> >
> > I meant the results from the user study will be helpful.
> > It is well known that most metrics for image synthesis models have some limitations such as FID, IS, and LPIPS.
> > So, the preference results from users (using a crowdsourcing platform such as Amazon MT) on the images generated from comparable models and the proposed method will help to validate the superiority of the proposed method. Of course, the authors can use their own user study in an objective setting instead of an open crowdsourcing platform.
> >
> > I will carefully read the response.

---

### Official Review · AnonReviewer3 · 2019-11-04
**Official Blind Review #3**

**Rating:** 3

**Review:**

This paper proposes an extension of the conditional GAN objective, where the generator conditions on an attention map produced by the discriminator in addition to the input image. The motivation is that the discriminator is usually too powerful, and so the gradient the generator receives is often too small in magnitude. By conditioning on the attention map, the generator could leverage information about the regions in the image that the discriminator attends to and use it to generate a new image that better fools the discriminator.

My main concern is about whether the proposed extension achieves the desired goal. The intuitive motivation provided in the paper aims to add a cooperative component to the two-player game, but the min-max objective corresponds to a zero-sum adversarial game. As a result, when training the discriminator, the discriminator is encouraged to reveal as little information as possible via the attention map, so that the loss maximized. This appears to be the opposite of the desired behavior, so the objective needs to be reformulated.

Also, it is unclear how inference is performed: at test time, the attention map is unknown and so some placeholder must be used in its place. The paper should clarify what is done at test time, and clearly state the shortcomings as a result of this, i.e. different procedures are used for training and testing, which is not principled. I imagine the generator could rely too much on the attention map as a result - how this is alleviated/prevented should be explained.

Figure 2: Only the qualitative results for unsupervised image-to-image translation are available; qualitative results for supervised image-to-image translation should also be provided.

While the quantitative improvement over existing methods is somewhat insignificant, I appreciate the authors discussing their hypotheses why this might be the case. It would be more useful to empirically validate these hypotheses as well. For example, for the claim that "maybe the attention map only focuses on a few domain specific classes so the generator works too hard on those classes and ignores others", it might be good to compute the average per-class attention map intensity to show that some classes appear rarely in the attention map.

The evaluation protocol should be explained in greater detail (perhaps in the appendix); the segmentation model (which I assume is FCN) should be described and each of the evaluation metrics (per-pixel acc., per-class acc. and IoU) should be described for the benefit of researchers outside the area.

pg. 4: "in their implementation contains several Resblock (He et al., 2016), which makes it infeasible in our framework". Why is it infeasible?
pg. 6: What are the architectures used by the baselines? Are they comparable to the architecture the proposed method used?

Minor Issues:

pg. 3: "differences between P_x and G_Y \cdot P_y, P_y and G_X \cdot P_x are minimized" - confusing; should rephrase as "the difference between P_x and G_Y \cdot P_y and the difference between P_y and G_X \cdot P_x are minimized". Also should replace \cdot with \circ.
pg. 4: "like random noisy" -> "like random noise"
pg. 4, last paragraph: "Our trainable attention module follows the same structure of the attention block in RAM (Wang et al., 2017). They built a very deep network with several such blocks, each containing two branches: mask branch and trunk branch. Mask branch cascades the input features through a bottom-up top- down architecture that mimics human attention. Trunk branch is applied as feature processing." - this is very confusing; it would be easier to refer readers to the appendix.
pg. 5 - "Attention mask can potentially break good property of the raw input." - what does this mean?
pg. 6 - "as showed in Table 4.1" -> "as shown in Table 4.1"


**Experience Assessment:**

I have published one or two papers in this area.

**Review Assessment: Checking Correctness Of Derivations And Theory:**

N/A

**Review Assessment: Checking Correctness Of Experiments:**

I assessed the sensibility of the experiments.

**Review Assessment: Thoroughness In Paper Reading:**

I read the paper thoroughly.

---

> ### Author Response · Authors · 2019-11-09
> **Response to Reviewer 3**
>
> Sorry for the late response and thank you for your constructive comments and demonstrated interest in the presented work, below we address each of your points in turn. 'G' denotes the generator and 'D' denotes the discriminator.
>
> (1). For the concern that D is encouraged to reveal little information via the attention map due to the nature of zero-sum game. Actually, D does not explicitly 'output' an attention map. Both attention mechanisms we used can be considered as an analysis process for D. More concrete, if D is dealing with a generated image $x$, the attention map is computed based on the activations of a given layer and it will be detached from the computation graph. Thus, this attention map only affects G's parameters, also the small component to compute Alpha Channel while using RGBA concatenation, and has no effect on D's parameters;
>
> (2). How inference is performed. There exists a loop between the G and D. That is, we need an attention map to synthesize an image while we also need a generated image to compute an attention map. An ALL-ONE attention map is used in the test phase. The intuition is the attention map highlights the crucial regions in the view of D. Since we don't have the attention map in the test phase, we can simply assume the D does the classification based on the whole image and G should pay attention to every pixel. Indeed, the procedures in the train and test phase have a little different, but G can still enjoy the benefits from such a training process. How we alleviate the dependence on the attention map is by using the proposed concatenation method. (Related to answer (9). So the attention map can only amplify the information but never hurt the original input); (We are trying to solve this difference between train/test in our future work, but it's out the scope of this paper)
>
> (3). The reason why We didn't provide the qualitative results for the supervised I2I translation are 1). The improvement for supervised translation is limited, based on the quantitative result and 2). The page limitation. We can provide the qualitative result for supervised translation in the revised paper but we are arguing is this result really necessary;
>
> (4). Empirical hypothesis justification. Thanks for your suggestion, we think the average per-class attention map intensity can help to justify our discussion and we will add those experiment results in the appendix in the later revision;
>
> (5). The evaluation protocol description. Sure, we will add the description of the FCN network and the metrics. But we haven't decided whether this part should go to the appendix or the main content;
>
> (6). Why the original RAM implementation is infeasible in our framework. Based on (Wang et al., 2017), they build a very heavy network (from 56 layers to 452 layers) to solve a complex image classification problem. However, in our case, G is already far behind D and D is conduct a relative easy binary classification, compared to the task in their paper. Moreover, using such a heavy and powerful discriminator will further break the balance between G and D. That's the reasons we decided to simplify its architecture. Maybe we should call it impractical;
>
> (7). The architecture used by the baselines. Basically, our unsupervised I2I model is based on the official implementation of CycleGAN and the supervised version is built upon the pix2pix's official implementation. We didn't change G's architecture, except the input channel is 4 when using RGBA concatenation. For the post hoc attention, we didn't change D's architecture since only the activation of a specific layer is required. For the trainable attention module, the modified discriminator (with mask & trunk branch) is powerful than the original one in the CycleGAN/pix2pix. We tried to replace the original discriminator with this modified D. But the performance of G decreases so we stay with the original one. This is expected because a more powerful D will further break the equilibrium between G and D. Both UNIT and StarGAN are more complex than CycleGAN so we think all of the baselines are comparable;
>
> (8). The confusing long sentence about RAM. We will rewrite this part and refer the reader to the appendix in the revision.
>
> (9). Attention mask can potentially break good property of the raw input. Suppose $x$ is the input image and $A_x$ is the attention map of it. We know that all the elements in $A_x$ are in [0, 1]. Using $X \times A_x$ instead of $x \times (1 + A_x)$ will decrease the value in the original image, and sometimes results in zero. However, 1) An attention map highlights important regions from the perspective of D. But a pixel with attention value 0 doesn't mean it's useless for the image translation task. By using residual multiplication, we give G the opportunity to bypass the effect of the soft attention mask. 2). $A_x$  is supposed to amplify the signal of the corresponding pixel;
>
> (10) Typos and other issues will be fixed in the future revision;

---

### Author Response · Authors · 2019-11-15
**Summary of Our Final Revision**

Dear Reviewers,

Thank you very much for your valuable comments and suggestion. Based on your’ suggestion and question, we did a massive modification to our paper draft. Below is a summary of our modification.

Minor:
We removed some unnecessary terms to make the paper more clear and compact;
To make the paper more consistent, we updated some terms and applied their abbreviations.
Residual Multiplication -> Residual Hadamard Production -> RHP
Trainable Attention Module -> TAM
Post hoc attention -> PHA

Major:
Sorry for the long update list, we completely reorganize and rewrite the experiment part.

We add the comparison with [Huh et al: Feedback Adversarial Learning: Spatial Feedback for Improving Generative Adversarial Networks] in the relative work. But unfortunately, we cannot reproduce their method since their Git Repo is down — Section: Relative Work
We add the explanation about the infeasible of RAM and why attention mask may break the good property of the raw input in the Method — Section: Method
We explained how inference is performed at test time at the beginning of our new Experiment section. — Section: Experiment
We added a new attention embedded GAN, called AGGAN, as a new baseline to address the novelty issue. By comparing with this baseline, we can show the difference between our method and previous attention embedded framework. — Section: Experiment
We moved the RAM description to the Appendix and shown how we avoid the problem caused by the zero-sum game between the generator and the discriminator — Appendix A
We added some extra statistic analysis with Cityscape dataset and empirically justified our hypotheses using attention map intensity — Appendix D
We explained the evaluation protocol in detail in the Appendix. — Appendix B
We tried a new challenging dataset ‘winter2summer’ and presented the experiment results.
We computed the KID score (Both target only and mean) for all datasets, and thanks for this metric, we can show that our method significantly outperforms that the baseline in the numeric view. — Section: Experiment
We added two large qualitative comparison plots with baselines and demonstrate the advantage of our method — Section: Experiment
We provided the experiment for all four network combination (Recall that we have two attention mechanism and two concatenation methods) — Section Experiment
We moved the qualitative result of Cityscape from the main body to Appendix —Appendix D

Finally, sorry for the massive update and thank you for your time and suggestion!

---

### Decision · Program_Chairs · 2019-12-19

**Decision:**

Reject

**Comment:**

The paper proposes to augment the conditional GAN discriminator with an attention mechanism, with the aim to  help the generator, in the context of image to image translation. The reviewers raise several issues in their reviews. One theoretical concern has to do with how the training of the attention mechanism (which seems to be collaborative) would interact with the minimax, zero-sum nature of a GAN objective; another with the discrepancy in how the attention map is used during training and testing. The experimental results were not significant enough, and the reviewers also recommend additional experiment results to clearly demonstrate the benefit of the method.